# Magnetite-Supported Gold Nanostars for the Uptake and SERS Detection of Tetracycline

**DOI:** 10.3390/nano9010031

**Published:** 2018-12-27

**Authors:** Paula C. Pinheiro, Sara Fateixa, Helena I. S. Nogueira, Tito Trindade

**Affiliations:** Department of Chemistry, CICECO-Aveiro Institute of Materials, University of Aveiro, Campus de Santiago, 3810-193 Aveiro, Portugal; pcpinheiro@ua.pt (P.C.P.); helenanogueira@ua.pt (H.I.S.N.); tito@ua.pt (T.T.)

**Keywords:** Au nanostars, magnetite nanoparticles, water pollutants, SERS, antibiotics

## Abstract

Magnetite nanoparticles (MNPs) decorated with gold nanostars (AuNSs) have been prepared by using a seed growth method without the addition of surfactants or colloidal stabilizers. The hybrid nanomaterials were investigated as adsorbents for the uptake of tetracycline (TC) from aqueous solutions and subsequent detection using surface-enhanced Raman scattering (SERS). Several parameters were investigated in order to optimize the performance of these hybrid platforms on the uptake and SERS detection of TC, including variable pH values and the effect of contact time on the removal of TC. The spatial distribution of TC and AuNS on the hybrid composites was accomplished by coupling SERS analysis with Raman imaging studies, allowing also for the determination of the detection limit for TC when dissolved in ultrapure water (10 nM) and in more complex aqueous matrices (1 μM). Attempts were also made to investigate the adsorption modes of the TC molecules at the surface of the metal NPs by taking into account the enhancement of the Raman bands in these different matrices.

## 1. Introduction

Antibiotics dissolved in natural waters are regarded as organic wastewater contaminants of general concern due to their potential risks to human health and ecosystems. The widespread use of antibiotics has increased the concerns related to the adverse contamination of waters, due to the impact on human health through direct and indirect consumption and also in the treatment of infections caused by antibiotic-resistant bacteria [1,2,3,4]. Antibiotics can be released into municipal wastewaters due to incomplete metabolism in humans or due to disposal of unused antibiotics acquired for human consumption and veterinary uses. Among the antibiotics that pose great concern as organic wastewater contaminants, there is tetracycline (TC), which is a widely used broad-spectrum antibiotic, although in the European Union it is restricted to the treatment of infectious diseases in humans and animals. Tetracyclines were first reported in the scientific literature in 1948 as natural products from the soil bacteria actinomycete [5,6]. According to the European surveillance of veterinary antimicrobial consumption, of the overall sales of antimicrobials in the 30 countries in 2015, tetracycline antibiotics represented 32.8% of the total sales for veterinary usage [7]. It has been reported that only a small portion of TC is metabolized in the organism and 70% is excreted via urine and feces in an unaltered form [8]. Indeed, the occurrence of TC has been reported in animal products and in aquatic environments from ng/L up to μg/L [3,8,9]. In Europe, the maximum residues limits for TC in different animal food categories are set by Commission Regulation (EU) 37/2010 [10]. As such, water monitoring methods that allows for the detection of vestigial TC in waters are of great relevance. 

Several instrumental techniques have been used for the detection of TC, including capillary electrophoresis [11], high-performance liquid chromatography [12,13], immunoassay [14], fluorescence-based assays [15], antibody-based electrochemical biosensors [16] and microbiological methods [17]. Although these techniques can achieve high sensitivity and selectivity, they are time-consuming and in some cases cost-intensive. Alternatively, surface-enhanced Raman scattering (SERS) spectroscopy is a sensitive vibrational technique that might allow for the trace detection of vestigial molecules adsorbed on metal surfaces, typically Au or Ag colloidal nanoparticles (NPs) [18,19,20,21,22]. SERS provides multiple advantages over other techniques that include the high sensitivity, spectroscopic fingerprint properties and simple sample preparation. Additionally, due to narrow well-resolved bands characteristic of inelastic light scattering, SERS can provide abundant vibrational information of the adsorbed molecules and be applied to multiplex detection. In fact, this technique has been explored for the TC detection in several types of samples [23,24,25,26,27,28,29,30,31]. For instance, Li et al. have applied Au hollow spheres to detect TC and the limit of detection reached 0.1 μg/L [31]. Dhakal et al. have reported a cyclodextrin-modified Ag colloid for on-site quantification of TC residues in whole milk, achieving a detection limit of 0.01 ppm, which is below the maximum residue limit (MRL) set by governmental authorities [27].

Although ultrasensitive analysis can be carried out with SERS, antibiotic compounds in natural waters are normally present in low concentration in such complex analytical matrices. An interesting possibility to circumvent this problem involves the capture of molecules of the pollutant by a sorbent material that besides allowing for pre-concentration procedures also acts as an SERS substrate for optical detection [32]. In this context, we have been interested in developing magneto-plasmonic hybrid nanomaterials that combine magnetic properties and SERS activity, thus exploring the magnetic separation ability offered by a magnetic oxide (e.g., Fe_3_O_4_) and the plasmonic behaviour of a nanometal (e.g., Au) [32,33]. Several approaches have been reported in the literature to produce magneto-plasmonic nanomaterials for SERS-sensitive detection, aiming at environmental applications [32,33,34,35,36,37,38,39,40,41,42,43,44,45,46,47], but only a few reports have described the SERS detection of antibiotics [33,41,44].

In this research, we have investigated a simple and low-cost surfactant-free seed growth method to produce gold nanostars (AuNSs) supported on magnetite NPs (MNP-AuNS) envisaging their use as nanosorbents for TC extraction and detection. These multifunctional nanostructures can be used as nanosorbents for the pre-concentration of the target analyte by magnetic separation as the first operational procedure followed by SERS detection of TC in the solid phase. This enrichment procedure and the presence of AuNS in the hybrid material contribute to the enhancement of the Raman signal of the molecular probe. Therefore, these hybrid materials were evaluated for the magnetic capture and SERS detection of TC in distinct aqueous matrices, which were spiked with this antibiotic. 

## 2. Results and Discussion

### 2.1. Characterization of MNP-AuNS 

Figure 1 illustrates the whole procedure for the fabrication of MNP-AuNS and their use in the uptake and detection of TC by SERS. At the outset, well-defined cubic-shape Fe_3_O_4_ NPs were first synthesized through partial oxidation of a ferrous salt in alkaline media, using KNO_3_ as the oxidant agent. The magnetic core ensured the convenient separation of the material from the aqueous solution and the Au nanostars at the MNP surfaces acted as SERS substrates. Hence, the MNP-AuNS nanostructures combined the SERS activity of Au NPs and the magnetic properties of Fe_3_O_4_ NPs. Briefly, a TC aqueous solution was mixed with MNP-AuNS hybrid substrates and then the magneto-plasmonic substrates were separated magnetically, using a magnet and washed with ultrapure water under magnetic confinement. The resultant MNP-AuNS particles with adsorbed TC were re-suspended in water and 10 μL of hybrid particles were dropped on a glass substrate placed over a NdFeB laboratorial magnet. After drying in air at room temperature, the SERS analysis was conducted, with laser excitation of the TC supported onto the hybrid substrates.

Figure 2a shows the TEM images of the MNPs, confirming that the Fe_3_O_4_ particles have a well-defined cubic shape and an average dimension of 80 nm (diagonal). Our previous work reports that the method employed results in cubic-shaped ferrimagnetic NPs with a saturation magnetization of 86 emu/g composed of magnetite as the main crystalline phase [33,48,49]. The surface of the as-prepared MNP was decorated with AuNS using an adaptation of the seed-mediated growth method described by Wang and co-workers [50]. Thus, Figure 2b shows Au nanospheres with an average size of 14.9 ± 3.0 nm covering the cubic-shaped Fe_3_O_4_ MNP. Then, these MNP-Au seeds were dispersed in an aqueous solution containing Au(III), and the solution was then acidified with HCl, followed by simultaneous slow addition of ascorbic acid and AgNO_3_. According to the literature, in these conditions, the presence of Ag^+^ plays an essential role in the growth mechanism to control the anisotropic growth of Au branches; ascorbic acid is responsible to reduce Au^3+^ to Au^0^; the acidic pH of the synthesis solution leads to more red-shifted plasmon bands because it favours the formation of larger nanostars; and the Au seeds control the AuNS size [50,51,52,53]. In the present work, the ratio between [Au^3+^] and [Au^0^] was set at 50, leading to the formation of AuNS with an average core size of 77.5 ± 20.1 nm and several branches, as can be seen in Figure 2c. An immediate advantage of using MNPs as substrates for the AuNS growth is that the particles can be isolated efficiently from the colloid through simple magnetic separation, although it should be noticed that the use of the seed-mediated growth method to obtain AuNS on the surfaces of MNP in an acidic medium probably resulted in some detachment of Au seeds from these substrates (see Appendix A). Nevertheless, it was found necessary to obtain the MNPs decorated with smaller AuNSs.

The optical spectra of the different colloids were measured and presented in the Appendix A. The addition of MNPs to the Au seeds red-shifted the localized surface plasmon resonance (LSPR) band of Au NPs and also resulted in band broadening. We attributed this behaviour to particle aggregation effects as demonstrated by SEM analysis (Figure 2b, left). Additionally, the broadening of the LSPR band for the AuNSs when interacting with the MNPs has been explained due to plasmon coupling, refractive index effects, particles’ heterogeneity and tips’ polydispersity. Other researchers have reported similar optical behaviour for other hybrid nanostructures, such as MNP-AuNS [54] and rGO-AuNS [50].

Figure 3 shows the powder XRD diffractograms of the Fe_3_O_4_, MNP-Au seeds and MNP-AuNS samples. The diffraction features that appear at 30°, 35.5°, 43°, 54°, 57.5° and 47.5° correspond to the (220), (311), (400), (422), (511), and (440) planes, respectively, of the inverse spinel structure of magnetite (JCPDS file no. 19-0629) (International Centre for Diffraction Data-Powder Diffraction File (ICDDPDF) No. 00-005-0566). The difractograms (a) and (b), corresponding to MNP-AuNS and MNP-Au seeds particles show four additional peaks assigned to the (111), (200), (220) and (311) planes of Au NPs with the face-centered cubic (fcc) structure (indicated with grey bars in Figure 3) (JCPDS Card No. 04-0784). The XRD patterns for MNP-AuNS particles present an additional peak at 34°, which matches with the (104) plane of hematite (JCPDS Card 33-0664). This small amount of hematite (α-Fe_2_O_3_) can be attributed to oxidation of surface iron sites due to reduction of Ag^+^ adsorbate ions during AuNS growth. 

### 2.2. Uptake of TC from Aqueous Solutions Using MNP-AuNS Nanosorbents

The molecular structure of TC comprises a four-fused-ring structure (Figure 4a), whereas ring D is aromatic but rings A, B, and C include saturated carbon centres. This three-dimensional architecture is almost planar (rings B, C and D are planar) including a kink between rings A and B (Figure 4b) [55]. Figure 4c shows the UV-Vis spectrum of TC aqueous solutions (10 μM), which presents a multiplet of bands with maximum absorption wavelengths peaked at 250 nm, 278 nm and 358 nm corresponding mainly to π→π* transitions.

The performance of the hybrid nanomaterials in the uptake of TC from aqueous solutions was assessed at variable pH values and for different contact times. First, the effect of the contact time on the adsorption capacity of MNP-AuNS, for the initial antibiotic concentration of 10 µM at pH 5–6, is shown in Figure 5. Pure Fe_3_O_4_ particles and the MNP-AuNS composite were dispersed in aqueous solutions of TC (10 μM) for 20 min and 24 h, respectively, and then the materials were recovered by magnetic separation from the solution. The amount of adsorbed TC was indirectly evaluated by UV-Vis measurements performed on the supernatants.

The adsorption of TC using Fe_3_O_4_ NPs reached around 15%, either after 20 min or 24 h. On the other hand, the MNP-AuNS particles adsorbed 31% of TC dissolved in the original solution, after 20 min and 24 h contact times. Control experiments carried out in parallel in the absence of sorbent particles under the same conditions of pH and contact time have demonstrated no losses of TC. Hence, the decrease of TC concentration in solution when contacted with sorbent particles was ascribed to adsorption phenomena. These results also suggest that the adsorption of TC had increased due to the presence of the Au nanostars at the surface of the magnetic particles.

TC is an amphoteric molecule with three functional groups that confer a marked pH-dependent behaviour on solubility [56]. Appendix A shows the molecular structure and pKa values for tetracycline (see Appendix A). TC presents an amphoteric behaviour due to the presence of ionisable functional groups and possesses three acidity constants. The first deprotonation involves the C1–C3 tricarbonyl methane (pKa = 3.3), leading to the zwitterionic form of the neutral compound. The second pKa (pKa = 7.7) is due to the deprotonation of the ketophenolic hydroxyl groups in C10 and C12. Above pH 9.6, TC exists in the aqueous solution as a divalent anion due to the neutralization of the C4-protonated dimethylamino group. This knowledge on the pKa constants gives an indication about the chemical species of TC in the solutions under analysis and is also relevant to inquire about the molecular forms that might interact with the colloidal particulates. In these regards, the surface charge of the Fe_3_O_4_ and MNP-AuNS is also important and the isoelectric points were found to be 5 and 4, respectively.

The influence of pH on the performance of the MNP-AuNS as sorbents for TC was studied in the pH range 2–10 for a contact time of 20 min (Figure 6). Control experiments were performed in the same conditions using neat Fe_3_O_4_ particles. The adsorption of TC onto MNP-AuNS increased at pH values between 4 and 6, above which the trend was reversed. As expected, in a strong acidic solution (pH < 3.3), the electrostatic repulsion between the cationic moieties of TC and positive surfaces of Fe_3_O_4_ and of MNP-AuNS led to low TC adsorption. When the pH of the suspension was higher than the isoelectric point of the sorbents (Fe_3_O_4_: pH 5; MNP-AuNS: pH 4), the negative charges on the surfaces of Fe_3_O_4_ and MNP-AuNS interact with cationic TC. Thus, the adsorption increased with increasing pH and reached a maximum at approximately pH 5 and pH 6, using Fe_3_O_4_ and MNP-AuNS, respectively, and then decreased with the further increase of pH. Indeed, under alkaline conditions, there was a reduction of adsorbed TC due to electrostatic repulsions between negatively charged TC and the adsorbents. Thus, the most favourable pH range for TC adsorption onto the MNP-AuNS sorbents was observed at pH 5–6. Although the adsorption of TC at such a pH window is the most favourable, it still occurs at other pH values due to the presence of other type of interactions between the TC adsorbates and the particles’ surfaces. In fact, the observed sorption behaviour can be explained by taking into account a combination of electrostatic interactions and covalent type/inner sphere-type interactions [57,58,59,60,61,62]. The latter explaining the adsorption of TC at low pH values, where both NPs and TC are positively charged, and also at pH > 7, where both the particles’ surface charge and TC species are negative.

### 2.3. SERS Detection of Tetracycline Using MNP-AuNS Substrates

Figure 7 shows the SERS spectrum of TC on MNP-AuNS substrates and the Raman spectrum of TC powder is also shown for the sake of comparison. Appendix A shows the Raman spectra of TC aqueous solutions (0.1 M and 10 μM) and MNP-AuNS powder. Note that MNP-AuNS originates low-background signal in the Raman spectrum and does not interfere with TC detection (Appendix A). The Raman spectrum of a TC aqueous solution (0.1 M) shows pronounced bands at 1619, 1447, 1315 and 1172 cm^−1^ (Appendix A), but these bands are not observed in more diluted conditions (10 μM) used in the SERS experiments. Additionally, no Raman bands were observed for TC of which aqueous solution (10 μM) was contacted with Fe_3_O_4_ NPs for 20 min (Appendix A). Similar experiments using colloid of Au nanostars (i.e., without magnetite) as SERS substrate did not provide any Raman signal for TC, which highlights the advantage in using a combination of the AuNS with the magnetite particles in the hybrid nanostructures. 

The SERS enhancement depends on several factors such as the morphology (size and shape) of the nanostructures, the excitation wavelength and the aggregation state of the nanoparticles as a mean for the formation of hotspots [21]. Some authors have also described off-resonant effects to explain the SERS effect for conditions, in which the laser wavelength is red-shifted in relation the LSPR band [63,64].

In our case, it is not obvious to consider an off-resonant effect due to the closeness energies of the excitation line to the LSPR band. In addition, the use of MNP-Au seeds as platforms did not show any Raman signal from adsorbed TC (Appendix A), which is consistent with a less favourable particle shape for hotspot formation (when compared to the stars). It has been reported that AuNSs have higher local electromagnetic fields on the tips due to the presence of hotspots at sharp edges of the NPs, thus producing strong Raman enhancement [65]. These observations are in line with our results.

The SERS spectrum of TC onto MNP-AuNS substrates shows most of the bands in accordance with the conventional Raman spectrum of TC powder but pronounced band broadening is also observed (Figure 7). In particular, the bands at 747, 888, 1347 and 1459 cm^−1^, which correspond to normal modes that involve the CO stretching from rings B, C and D, are clearly observed (Table 1). In addition, most of the bands in the SERS spectrum are shifted in comparison to those observed in the Raman spectra of TC in powder and TC in the aqueous solution (0.1 M). In SERS, charge transfer is a short-range mechanism that occurs when the molecular analyte is chemisorbed at the surface of a metal [67,68]. Although in this case the TC molecules are retained onto the metal surface, charge transfer also results in the weakening or deformation of molecular bonds. The dissociation of the O–H bond and formation of the O–Au bond result in a more pronounced Raman shift for the deformation-coupled C–O stretching modes of rings B, C and D. The absence of the Raman band at 1619 cm^−1^ assigned to vibrational modes that involve both the amide-NH_2_ moiety and the secondary amine of the dimethylamino group suggests that these groups of TC are not directly interacting with the AuNS surfaces. In fact, as discussed above, this is in accordance with the assumption that the protonated dimethylamino groups (pH 5) interact preferentially with negative surface moieties of the iron oxide via electrostatic interactions, as schematically illustrated in Figure 8. However, the sorption attachment of TC onto MNPs’ surface, involving π-cation bonding between Fe (II and III) ions and aromatic π-electrons of TC, cannot be discarded [69].

Figure 8 relates to the TC molecules directly attached to the particles’ surfaces (first layer). Additionally, the formation of overstacked TC assemblies, due to π–π stacking interactions between neutral H_3_TC molecules, can also explain the higher adsorption of TC at this pH window. This is consistent with the observation of strong SERS signals of the aromatic C=C backbone [70].

Raman spectroscopy coupled with imaging methods combines both spectral and spatial information and thus allows for the localized identification of TC species in the substrate. In order to explore the spatial distribution and spectral information of the TC molecules onto MNP-AuNS material, 2D surface Raman imaging was acquired for the best SERS operational conditions (Figure 9). High-resolution Raman imaging was performed by raster-scanning the laser beam over a surface area of 400 μm^2^ and accumulating a full Raman spectrum at each pixel (in total 22500 spectra), with a spatial resolution of approximately 0.13 μm. The integration of the absolute area underneath the TC characteristic band at 1347 cm^−1^ was used to establish the colour intensity scale in the Raman map. Thus, the brighter colours correspond to a higher Raman band intensity of TC and consequently its location in the MNP-AuNS surface. 

As compared to metal colloidal nanostructures used in SERS studies, such as Ag NPs or Au NPs, the use of solids poses issues concerning the homogeneity of the substrates. In order to make sure that the observed Raman signals are ascribed to an SERS effect due to the presence of the AuNS, all the measurements were performed by using Fe_3_O_4_ particles as the substrate in control experiments (spectrum presented in Appendix A). Moreover, the spectra were considered as representative after analysing several areas of the substrate using Raman mapping and also by randomly selecting points on the substrate and recording the corresponding SERS spectra (Figure 9b). Although these results show slight changes in the Raman intensities, it is clear that at this scale the SERS signals are always observed, regardless of the region analysed.

Note that the tips of the AuNS provide strong localized electromagnetic field (hotspots) in the hybrid substrates. Thus, it is possible that the application of the magnetic gradient on the substrates could provide an arrangement of the MNP-AuNS particles that also originates regions of strong Raman signal enhancement. This point was investigated in more detail by comparing the SERS sensitivity of adsorbed TC on MNP-AuNS before and after the magnetic gradient. In more detail, the sample was deposited on a glass slide (Appendix A) and the Raman spectrum acquired, then a NdFeB laboratorial magnet was placed below the glass side to induce an external magnetic gradient (Appendix A) and the Raman spectrum acquired. In this case, no significant changes were observed in the Raman bands of TC presented on the collected SERS spectra but a better signal-to-noise ratio was obtained for the sample submitted to magnetic gradient (Appendix A). This result might indicate creation of hotspots induced by aggregation due to the external magnetic gradient.

In order to assess the SERS sensitivity of the MNP-AuNS for TC, samples of aqueous solutions with variable concentrations of antibiotic have been analysed (10 μM to 10 nM, pH 5). Figure 10 shows the Raman spectra of TC obtained by the variation of the concentration, using MNP-AuNS as substrate and the correspondent Raman map by monitoring the band intensity at 1347 cm^−1^. The TC seems localized in preferential regions of the scanned area, which is more clearly observed for samples of lower TC concentration. As the concentration of TC increased, the antibiotic content became more homogenously distributed on the surface of the substrate with increasing brighter areas in the scanned area. Figure 10d shows that in these conditions the SERS intensity of the TC bands tended to decrease as the TC concentration decreased from 10 μM to 10 nM. It is noteworthy that the SERS signal for TC was observed for a concentration as low as 10 nM. This result demonstrates that Raman mapping combined with SERS analysis offers a powerful technique to detect vestigial TC in spiked water by using the hybrid nanomaterials described here. 

### 2.4. SERS Detection of Tetracycline in More Complex Spiked Aqueous Matrices

The vast majority of the studies reporting the SERS detection of antibiotics in water were carried out using ultrapure water. However, salted waters are the last receptors of such pollutants and the source of water used in many aquacultures. In order to investigate the reliability of these SERS substrates using natural waters, the SERS detection of TC was performed using more complex spiked aqueous samples using estuarine water from the Aveiro lagoon (A) and mineral water from Serra do Buçaco (B). Thus, a series of spiked waters with different TC concentrations have been contacted with MNP-AuNS, using experimental conditions similar to those described above. Figure 11 presents Raman images corresponding to the SERS spectra of TC adsorbed on MNP-AuNS particles that have contacted with natural waters A and B. The SERS images show the spatial distribution of TC molecules on the MNP-AuNS substrate, in which the brighter areas correspond to an increase of the Raman signal of the TC molecules due to the presence of active Raman scattering sites. The SERS spectra in Figure 11c are single Raman spectra taken from the bright spots observed in the respective Raman images of Figure 11a,b (1 acquisition, 0.1 s). The Raman spectra of the MNP-AuNS substrate with (A) estuarine water from Aveiro lagoon (salty water) and (B) mineral water are presented in the Appendix A for comparative purposes.

This result demonstrates that the MNP-AuNS nanosorbents reported here are effective for the uptake and subsequently SERS detection of TC dissolved in more complex matrices, such as salted waters. In fact, the Raman spectra acquired from the collected maps using both water samples (A and B) are similar to the Raman spectrum of TC shown in Figure 7b. It should be noticed that for the experiment with mineral water sample, an additional Raman band is observed at 1619 cm^−1^ in the SERS spectrum of TC (Figure 11c, spectrum B), which is assigned to the amide stretching and bending modes in ring A. A possible explanation for this observation is related to the presence of competing cations not only for surface sites of magnetite but also for TC complexation [71], limiting the interaction of TC molecules with the iron oxide and thus leaving amide groups free to interact with the SERS-active Au surfaces. The detection limit of TC in both salted water samples was 1 μM (Appendix A), thus being inferior to the value obtained for the laboratorial samples but demonstrating that the MNP-AuNS nanomaterials are effective for SERS detection of TC detection in more complex aqueous matrices. 

## 3. Materials and Methods

### 3.1. Materials

The following chemicals were used as purchased: ferrous sulfate heptahydrate (FeSO_4_·7H_2_O, >99%, Panreac, Barcelona, Spain), potassium nitrate (KNO_3_, >99%, Sigma-Aldrich, St. Louis, MO, USA), sodium citrate tribasic dihydrate (Na_3_C_6_H_5_O_7_·2H_2_O, 99%, Sigma-Aldrich), chloroauric acid trihydrate (HAuCl_4_.3H_2_O, ≥99.9%, Sigma-Aldrich) and potassium hydroxide (KOH, >86%, Sigma-Aldrich), ascorbic acid (C_6_H_8_O_6_, J. M. Vaz Pereira, Lisboa, Portugal), silver nitrate (AgNO_3_, 99.9%, Sigma-Aldrich), acid chloride (HCl, 37%, AnaloR Normapur, VWR International, Radnor, PA, USA), and tetracycline hydrocloride (TC, C_22_H_24_N_2_O_8_, Sigma-Aldrich). All the solutions were freshly prepared using ultrapure water (18.2 mΩ·cm).

### 3.2. Synthesis of Fe_3_O_4_ Nanoparticles

Magnetite particles were synthesized by the hydrolysis of FeSO_4_·7H_2_O, as described elsewhere [49]. Briefly, a solution containing 20 g of FeSO_4_·7H_2_O in 140 mL of deionized water previously flushed with N_2_, was boiled at 90 °C. Then, a solution of 1.62 g KNO_3_ and 11.23 g KOH in 60 mL H_2_O was added dropwise for 5 min to the Fe (II) solution, under nitrogen bubbling. The black powder obtained was stirred over 1 h at 90 °C, left overnight and then washed with water and separated magnetically.

### 3.3. Synthesis of MNP-Au Seeds

The MNP-Au seeds were synthesized by a modified method described elsewhere [50]. Briefly, cubic magnetite nanoparticles (60 mg) were dispersed in ultrapure water (50 mL) and then the solution was sonicated in the ultra-sound bath for 10 minutes, and then left immersed in an ice bath, over 15 min under sonication (horn Sonics, Vibracell, Newtown, CT, USA). A HAuCl_4_ solution (80 μL, 158.3 mM) was added and the mixture was mechanically stirred for 30 min at 500 rpm. Then, the solution was heated to 90 °C, and 500 μL of sodium citrate (1.0 M) was added dropwise. The above boiling mixture was stirred for 1 hour, forming MNP-Au colloidal seed particles. 

### 3.4. Synthesis of MNP-AuNS Nanocomposite

The MNP-AuNS nanocomposites were synthesized by a seed growth method [50]. Briefly, 200 μL of the as-prepared MNP-Au seeds colloid was added to 10 mL of a 0.25 mM HAuCl_4_ solution, followed by 5 μL of 1 M HCl at room temperature and mixed by gentle inversion for 5 seconds. Then, 100 μL of silver nitrate (8 mM) and 50 μL of ascorbic acid (100 mM) were simultaneously added. The solution was mixed by gentle inversion for 20 seconds as its colour rapidly turned from light yellow to blue. The MNP-AuNS nanocomposites particles were magnetically separated from the colloidal solution and dispersed in ultrapure water.

### 3.5. Adsorption Experiments of TC onto MNP-AuNS

The amount of adsorbed TC at the MNP-AuNS particles was determined by the interpolation of UV absorption of the supernatant at 358 nm. The performance of the hybrid particles was investigated for the same antibiotic concentration (10 μM) at pH 5–6, with 20 min and 24 h contact times. Hence, 10 mL of TC solution was added to 0.25 mg of MNP-AuNS. These mixtures were then incubated using a mini-rotor at room temperature. The influence of the pH solution on TC adsorption was evaluated at variable pH values, from 2 to 11. Generally, the pH of a TC solution (10 mL, 10 μM) was adjusted to the required value by adding HCl or NaOH. Then, 0.25 mg of MNP-AuNS were added and dispersed, and the batch experiment was run in the conditions described above, for a period of 20 min. The final concentration of TC was determined by UV-Vis spectroscopy. The pH of the solution was also measured at the end of the adsorption experiment to confirm that no pH significant variations arise from the addition of the MNPs. 

### 3.6. SERS Measurements

The SERS study was performed by adding the as-synthesized particles (0.25 mg) to aqueous solutions of TC (10 mL) with different concentrations (10 μM–10 nM). The mixtures were incubated for 20 minutes, using a mini-rotor, at room temperature to allow for the adsorption of TC to the surfaces of AuNS. The samples were then magnetically separated from the solution using a magnet and the MNP-AuNS particles were washed twice with ultrapure water. Particles were transferred to glass sides for SERS analysis and characterization and dried at room temperature. For all the SERS measurements, the pure magnetite particles were also used as the control sample. SERS measurements have been performed in different areas of the nanocomposites in order to check the reproducibility of the measurements. In order to assess the performance of the MNP-AuNS under ionic competition conditions, a more complex aqueous samples spiked with TC was also analysed: estuarine water from Aveiro lagoon (seawater) and mineral water from Serra do Buçaco. We have analysed five different samples (MNP-AuNS), in which we have used different MNP-Au seeds and different MNP-AuNS batch for the detection of TC in ultrapure water, three samples for estuarine water from Aveiro lagoon and two samples for mineral water from Serra do Buçaco. 

Raman images obtained using the WITec Alpha 300 RA+ were produced by raster-scanning the laser beam over a surface area of 20 × 20 μm^2^ and accumulating a full Raman spectrum at each pixel (in total 22500 spectra), with a spatial resolution of approximately 0.13 μm. The integration of the absolute area underneath the TC characteristic band at 1347 cm^−1^ was used to establish the colour intensity scale in the Raman map. A laser wavelength of 633 nm with a laser power of 0.2 mW was used. A 100× objective was used to view samples, and the integration time for each spectrum was 0.1 s. The time required to create Raman images using each integration time was 44 min. All the Raman spectra were data treated with background subtraction using the Project 5 programme that are available with the confocal Raman microscope (WITec, Ulm, Germany).

### 3.7. Instrumentation

The TC concentration in the supernatant was determined spectrophotometrically by monitoring the absorbance at 358 nm spectrophotometrically using a Jasco U-560 UV-Vis spectrophotometer (Jasco Inc., Easton, MD, USA). Linear calibration was used for quantification based on the curves between the concentration and peak intensity of a known standard of TC. TEM was carried out on a Hitachi H-9000 TEM microscope (Hitachi, Tokyo, Japan) operating at 300 kV. The TEM samples were prepared by placing a drop of the diluted colloid on a carbon-coated copper grid and the solvent was left to evaporate in air. The XRD data were collected using a PAN analytical Empyrean X-ray diffractometer (PANanalytical, Almelo, The Netherlands) equipped with Cu Kα. The XRD data were collected using a PAN analytical Empyrean Xan aliquot of the aqueous suspension of the hybrid nanostructure on a silicon holder. Raman spectral imaging was performed in a combined Raman–AFM–SNOM confocal microscope WITec alpha300 RAS+ at CICECO (WITec, Ulm, Germany), in the Chemistry Department of the University of Aveiro. A Nd:YAG laser operated at 633 nm. 

## 4. Conclusions

In conclusion, we have developed new hybrid nanomaterials comprising plasmonic nanoparticles (Au nanostars) supported on magnetic solid (Fe_3_O_4_ particles). These hybrid nanostructures were investigated as colloidal nanosorbents for subsequent SERS analysis of tetracycline in water samples. This research has demonstrated that this type of composite nanostructures shows great potential for water monitoring and purification, including the application to more complex aqueous matrices, containing a number of organic and inorganic interfering species such as salted waters. We anticipate that this type of hybrid nanomaterials can be applied to the analysis of TC in other types of fluids, provided the adequate sample preparation is performed. Overall, the MNP-AuNS particles can be regarded as multifunctional platforms for vestigial analysis, biological detection and environmental monitoring, due to their ability for pollutant uptake using magnetic separation and subsequent detection using Raman methods.

## Figures and Tables

**Figure 1 nanomaterials-09-00031-f001:**
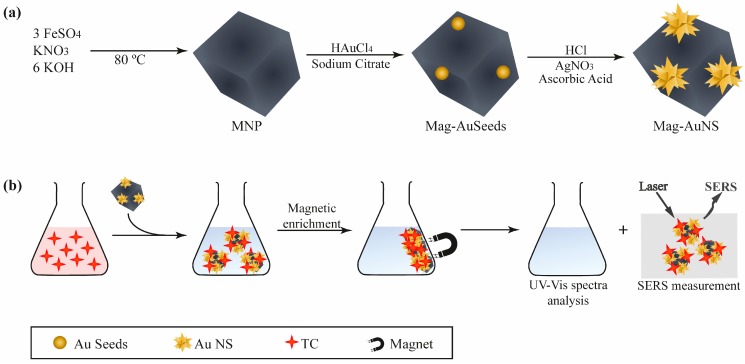
Schematic illustration of the (**a**) preparation of magnetite-gold nanostars (MNP-AuNS) hybrid materials; and (**b**) operating procedure for the magnetic uptake and surface-enhanced Raman scattering (SERS) detection of tetracycline (TC), using MNP-AuNS.

**Figure 2 nanomaterials-09-00031-f002:**
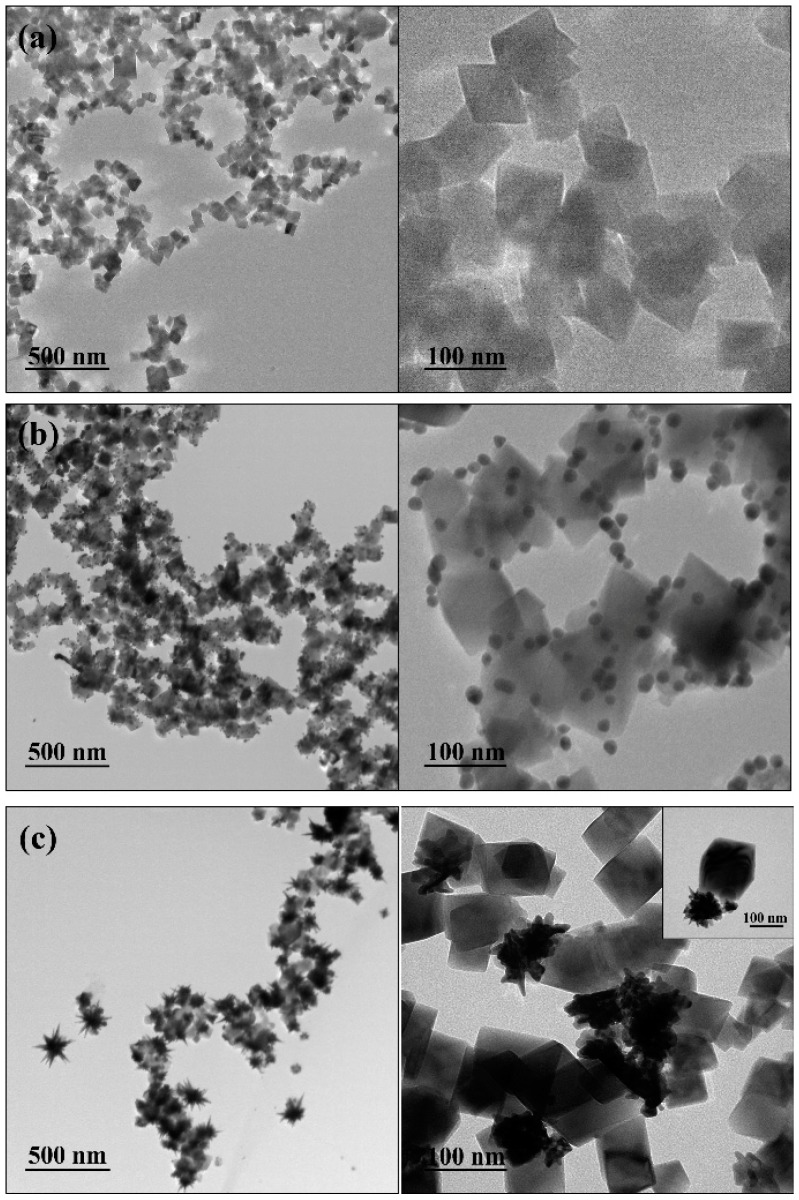
TEM images of (**a**) cubic Fe_3_O_4_ NPs; (**b**) MNP-Au seeds; and (**c**) MNP-AuNS (inset shows a high-magnification image).

**Figure 3 nanomaterials-09-00031-f003:**
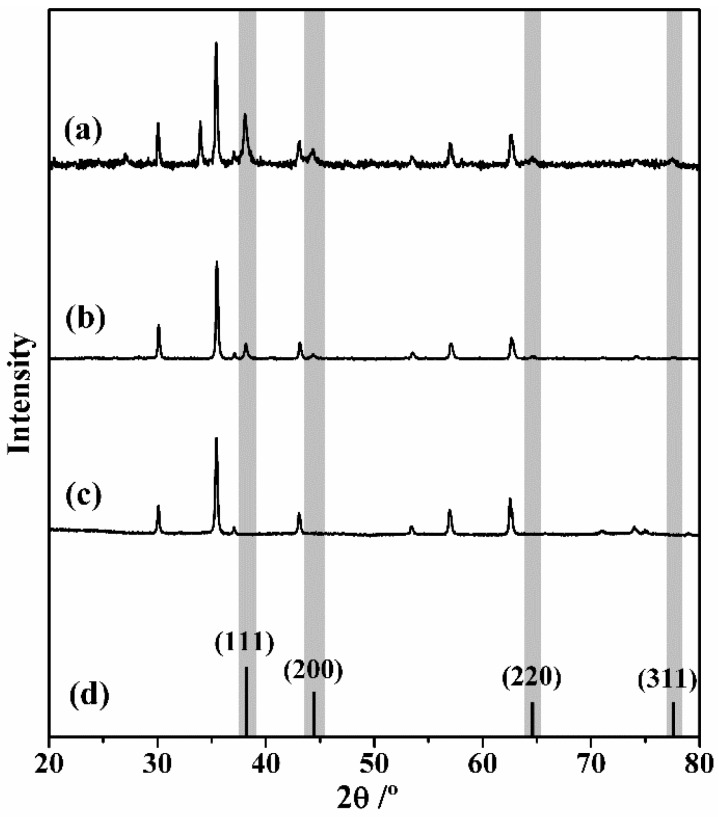
Powder XRD diffraction patterns of (**a**) MNP-AuNS NPs; (**b**) MNP-Au seeds; (**c**); Fe_3_O_4_ NPs; and (**d**) reported reflexions for crystalline gold with fcc structure (JCPDS Card No. 04-0784).

**Figure 4 nanomaterials-09-00031-f004:**
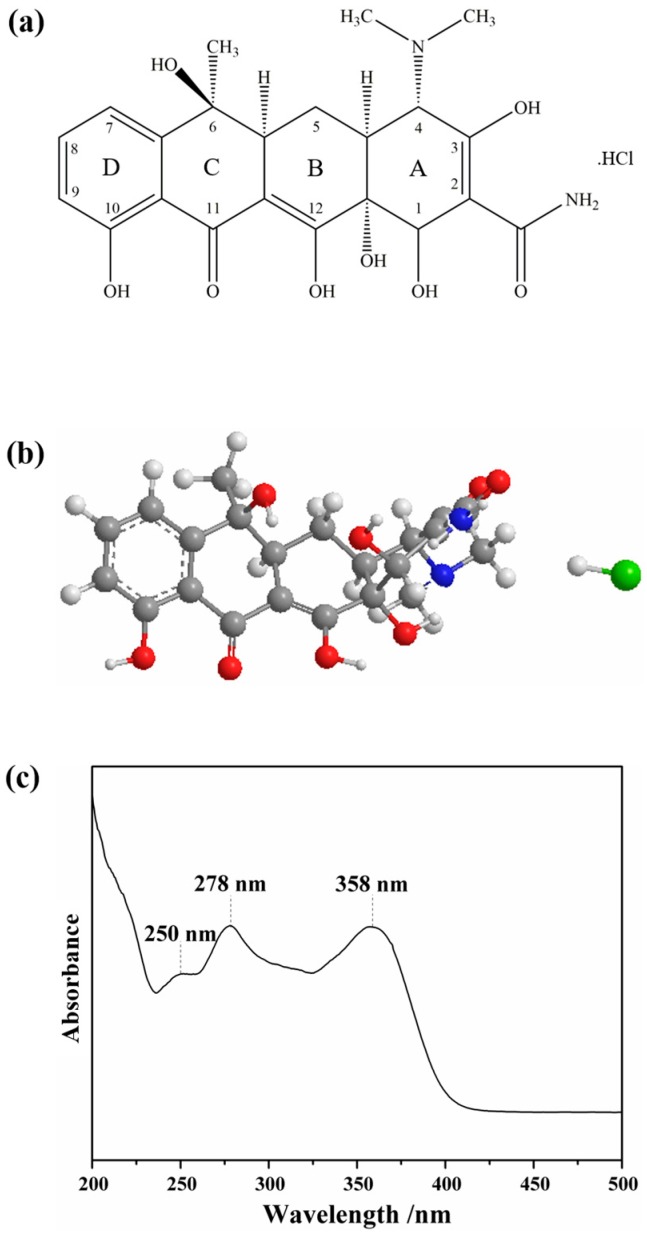
(**a**) Structural formula and (**b**) ball-and-stick representation of a TC molecule; and (**c**) visible spectrum of an aqueous solution of TC.

**Figure 5 nanomaterials-09-00031-f005:**
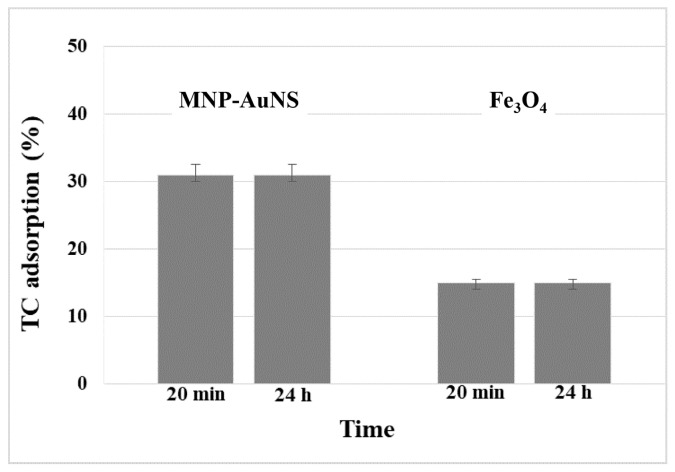
Amount of TC adsorbed on Fe_3_O_4_ NPs and MNP-AuNS, for an initial concentration of 10 μM in the antibiotic.

**Figure 6 nanomaterials-09-00031-f006:**
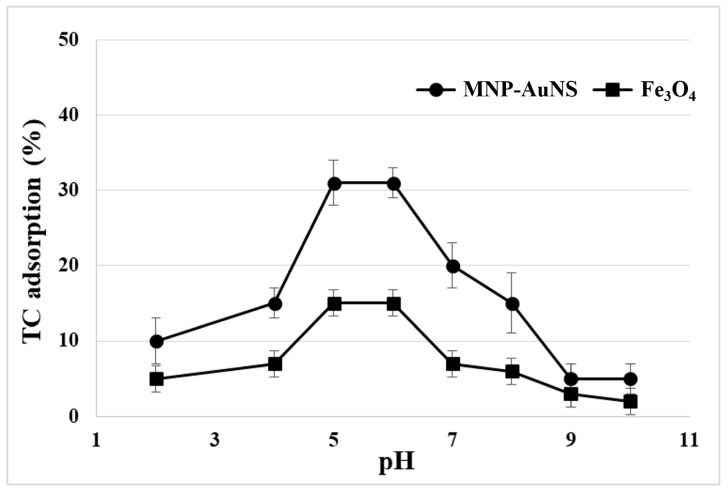
Effect of pH of the medium on the TC uptake, for an initial TC concentration of 10 μM and a contact time of 20 min.

**Figure 7 nanomaterials-09-00031-f007:**
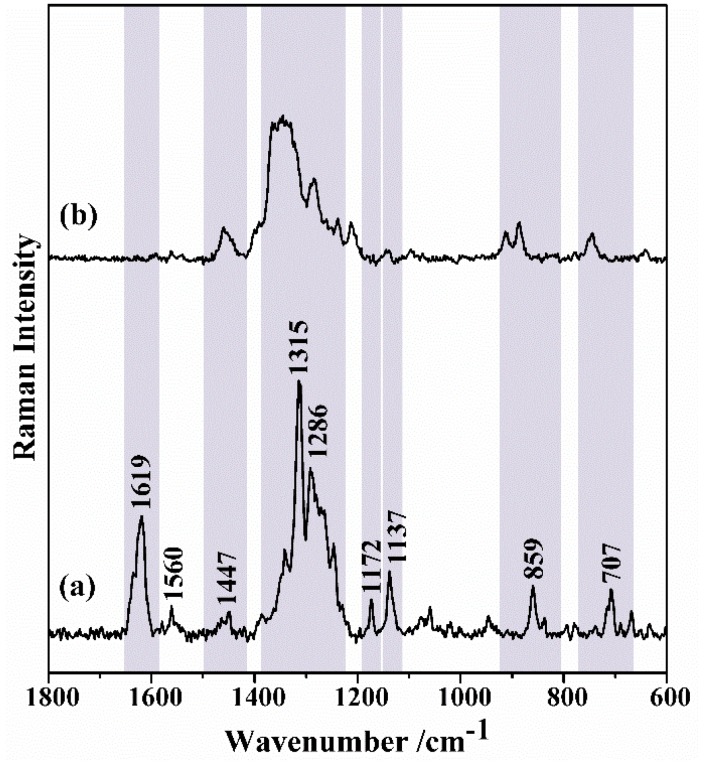
(**a**) Raman spectrum of tetracycline powder; (**b**) SERS spectrum of TC onto MNP-AuNS after contacting this sorbent with a solution of TC (10 μM). Grey bars highlight Raman bands observed for both the TC powder and TC adsorbed onto MNP-AuNS.

**Figure 8 nanomaterials-09-00031-f008:**
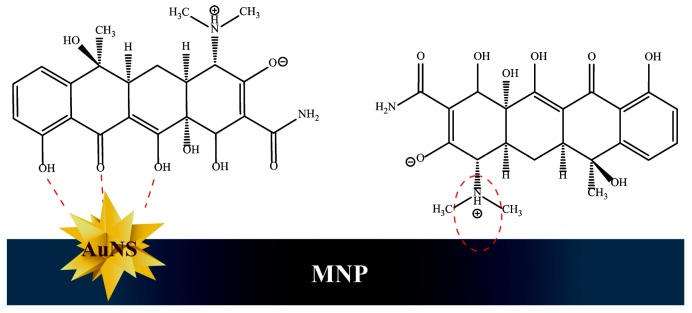
Schematic illustration showing TC molecules interacting with the hybrid nanostructures composed of AuNS and Fe_3_O_4_ particles.

**Figure 9 nanomaterials-09-00031-f009:**
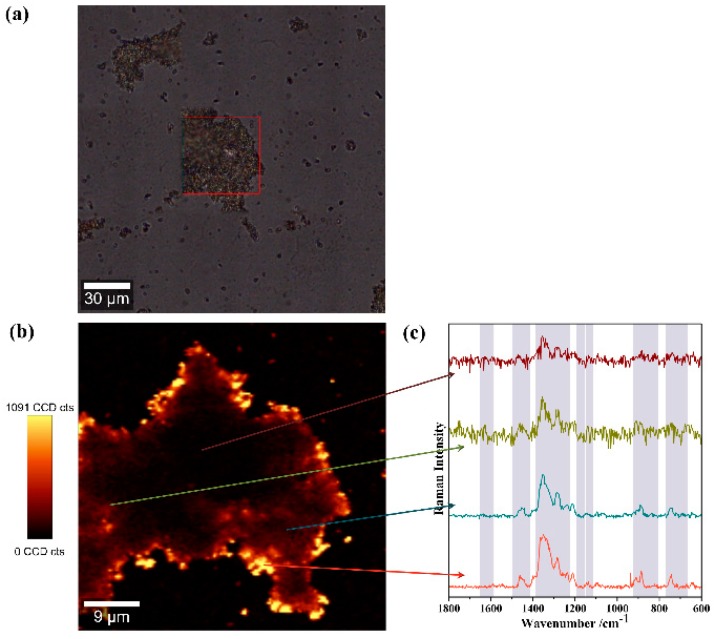
(**a**) Optical image of the MNP-AuNS hybrid substrates after contact with TC, with the Raman scanned area marked in red; (**b**) Raman image obtained using the integrated intensity of the Raman band at 1347 cm^−1^ in the SERS spectrum of TC (10 μM) using the MNP-AuNS as substrates; and (**c**) selected SERS spectra of TC collected at different points as shown by the arrows. The vertical bar shows the colour profile with the relative intensity scale.

**Figure 10 nanomaterials-09-00031-f010:**
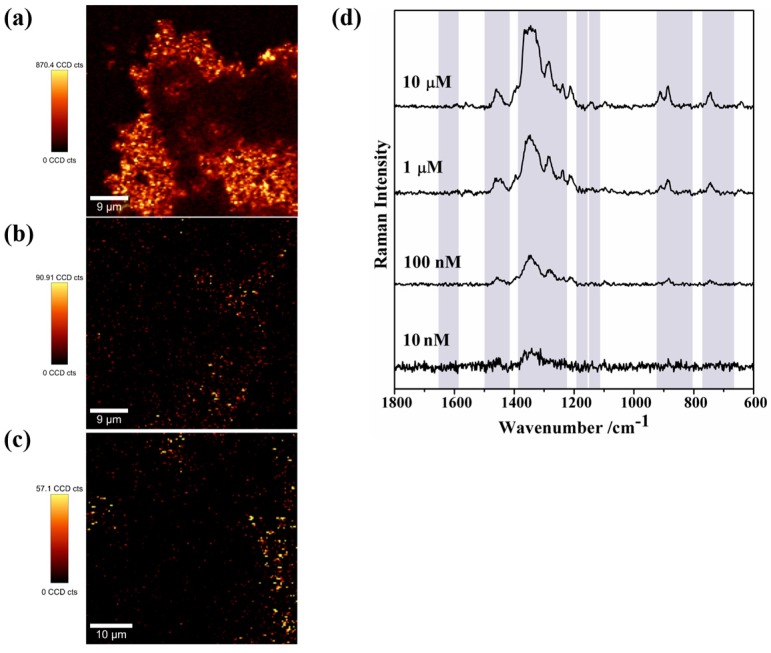
Raman images obtained with the integrated intensity of the band at 1347 cm^−1^ of TC recorded using MNP-AuNS as SERS substrate using different TC concentrations: (**a**) 1 μM; (**b**) 100 nM; and (**c**) 10 nM. The vertical bar shows the colour profile in each image, with the relative intensity scale. (**d**) Single SERS spectra of TC taken from the Raman images using different TC concentrations.

**Figure 11 nanomaterials-09-00031-f011:**
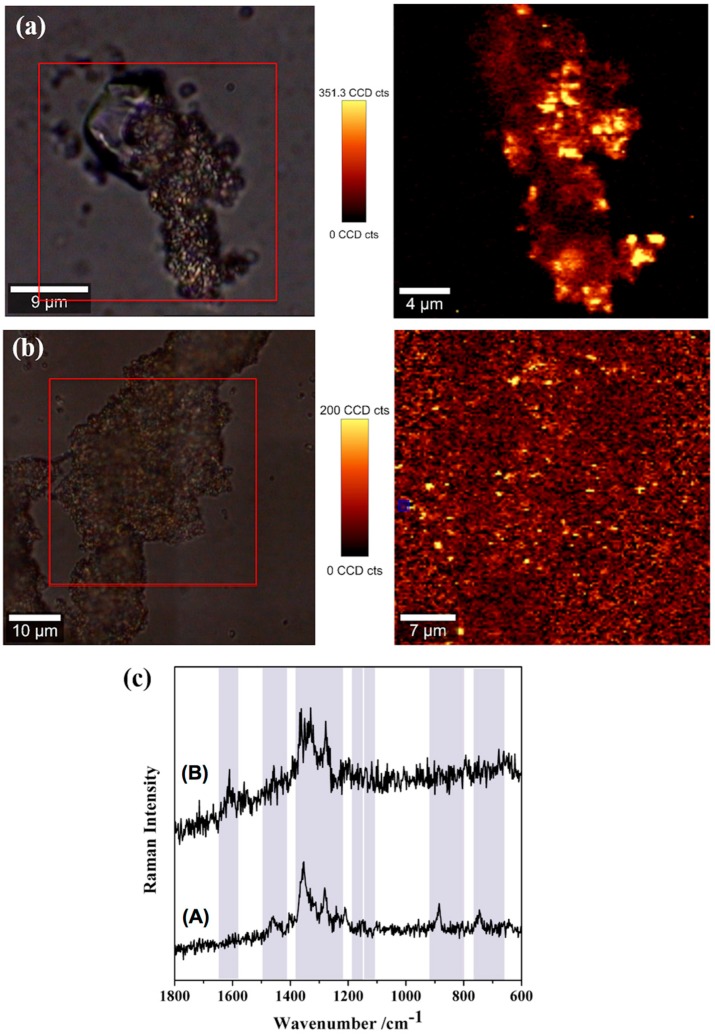
Raman images obtained using the integrated intensity of the band at 1347 cm^−1^ in the SERS spectra of TC (10 μM) recorded using the MNP-AuNS as substrates (excitation at 633 nm, 0.2 mW laser power) for the TC spiked samples: (**a**) estuarine water from Aveiro lagoon and (**b**) mineral water. The respective optical images of the samples are on the left side, with the scanned area marked in red. The vertical bar shows the colour profile in each image, with the relative intensity scale; and (**c**) single SERS spectrum for TC (10 μM) using MNP-AuNS obtained from the Raman images above for (A) estuarine water from Aveiro lagoon and (B) mineral water.

**Table 1 nanomaterials-09-00031-t001:** Raman bands and respective assignments for the TC powder and TC adsorbed onto MNP-AuNS used as SERS substrates [24,25,27,28,29,55,66].

Raman (cm^−1^)	SERS (cm^−1^)	Assignment
1619	-----	υ(CO1) + υ(amid-CO) + δ(amid-NH) + υ(CO3) + υ(C2-C3) + υ(OH10,12) + υ(amid-CO2)
1560	-----	(C11O) + (CC)D
1447	1459	δ(CH7,8,9) + δ(OH12) + υ(D) + υ(CO10,CO11,CO12) + δ(amin-CH_3_) + δ(CH_3_6)
1315	1347	δ(OH10,12) + δ(CH4,4a,5,5a) + υ(C5aC11a) + υ(C1C2) + υ(C9C10,C10C10a,C10aC11) + υ(CO11,12) + υ(CO3) + δ(CH7,8,9)
1286	1283	δ(CH4,4a,5a) + δ(OH12) + δ(amid-NH) + υ(CO10) + υ(CO3) + υ(CH7,8.9) + υ(amid-NC) + υ(C4aC5) + υ(D)
1172	1175	υ(CO3)
1137	1133	υ(CO6) + υ(CO12)
859	888	δ(CC)A,B,C,D + υ(C15 O25 )
707	747	ω(amid-NH), v(CO6,12), γ(OH3,6,10,12,12a)

υ = stretching modes; δ = bending mode; ω, out-of-plane swing; γ, out-of-plane bending.

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
