# Peer review of "Magnetite-Supported Gold Nanostars for the Uptake and SERS Detection of Tetracycline"

_nanomaterials, 2018, doi:10.3390/nano9010031_

Reviewer 1 Report

The authors present the interesting concept of magnetite-supported gold nanostars for SERS sensing. It was found that the SERS enhancement was not negatively affected by the magnetite support. The SERS enhancement is not surprising and has been reported quite frequently before for solution-based nanoparticles and substrate-supported systems. The major weakness of the work lies in the lack of optical characterization. The hybridization can be expected to influence the LSPR of the Au nanostar, which might constructively or destructively interfere with the hot spot formation at the chosen excitation laser wavelength. After a careful comparison with related works of literature, the originality seems low without an in-depth discussion of the effect of the magnetite support on the nanostar plasmonics. I suggest that the authors try to revise the manuscript by explaining their distinction and the originality of their work. In general, the manuscript fits very well into the field covered by the journal Nanomaterials. However, the current state of the manuscript is below the journal standard. Consequently, I recommend publication in Nanomaterials after major revision. In addition, few comments might help to address some open questions, see below.

Comments:

1)     Fig 1b shows a sketch of the SERS effect. The shift from red to green light corresponds to anti-Stokes scattering. However, I assume that the authors did in fact measure Stokes scattering. Thus, the energy of the Raman scattered light should lower than the excitation line. Please revise.

2)     What is the influence of MNP on the AuNS plasmonics? At least the refractive index can be expected to significantly shift the plasmon resonance towards the red. Also, the anisotropic growth of stars at the interface might influence the LSPR. This needs to be discussed briefly. 

3)     The coverage of seeds and stars on MNPs needs to be discussed. First, the coverage seems quite high (Fig2b) for the seeds, then, for the grown AuNS it seems rather low (Fig2c). The authors might want to discuss the effect of nanostar loss/detachment from the support.

4)     How stable are the AuNS attached to the MNP? What physicochemical interaction drives the attachment?

5)     What is the reason that the highest TC adsorption was present around pH 5-6? Can this be understood?

6)     Fig 8 gives two possible adsorption scenarios of TC an AuNS and MNP. I suppose that pi-pi interactions might as well guide the adsorption (compare J. Phys. Chem. C 2015, 119, 9513−9523). This could explain the strong SERS signals of the aromatic C=C backbone.

7)     Fig. 9: Why is the SERS signal intensity within the bulk almost close to zero? It seems like that SERS enhancement happens exclusively at the borders. Why?

8)     The term "real sample" is misleading and confusing. Please paraphrase. 

9)     What is the reproducibility of the reported signals? How was the data treated (background correction)? Please revise and give more information about the corrected data and errors. For sensor and analytical applications, the responsiveness and robustness of a sensing device are essential for reproducibility and reliability. This needs to be discussed in the context of the presented material (please compare Aitchison et al. Faraday Discuss., 2017,205, 561-600).

10)  Citrate was used as a stabilizing agent in the dispersion. Can traces of citrate be detected in the SERS samples and which signals would be expected to appear for citrate?

11)  Typo: L.282 the scan area should be in µm2.

12)  The authors do not show any optical properties of their substrates. The UV-vis spectra of the MNP, MNP+seeds, and MNP+AuNS should be should and discussed. Also in regard to the choice of laser line for excitation (633nm). This needs to be discussed more thoroughly in the context of the origin of SERS (Faraday Discuss., 2017,205, 173-211.) 

13)  Does the laser line coincide with the LSPR peak? If not, this could be explained by off-resonant enhancement. Compare "Plasmonics in Sensing: From Colorimetry to SERS Analytics", DOI: 10.5772/intechopen.79055.

Author Response

Dear Reviewer, 

you can find the reply to the comments in the attachment.

Best regards

Sara Fateixa

Reviewer 2 Report

see attached file

Author Response

Dear Reviewer, 

you can find the reply to the comments in the attachment.

Best regards

Sara Fateixa

Round  2

Reviewer 1 Report

The authors answered all of the posed comments. However, not all issues have been properly resolved and some open questions remain. Especially, a possible mix-up of data needs to be addressed. The top two spectra plotted in Fig. 10d (10µM and 1µM TC) almost perfectly match the spectra of Fig. S5 (both 10µM before/after magnetic concentration). In the light of these inconsistencies, I cannot recommend publication in Nanomaterials before unless further revisions. Below, I repeat comment #13 which was not answered satisfactorily.

Comments:

1) In the earlier report, the authors were asked to comment on the excitation conditions of their samples and the effect of off-resonance conditions. "Does the laser line coincide with the LSPR peak? If not, this could be explained by off-resonant enhancement. Compare "Plasmonics in Sensing: From Colorimetry to SERS Analytics", DOI: 10.5772/intechopen.79055." The authors replied "Please see above", which is not answering the open question.

2) A possible mix-up of data needs to be addressed. The top two spectra plotted in Fig. 10d (10µM and 1µM TC) almost perfectly match the spectra of Fig. S5 (both 10µM before/after magnetic concentration), including identical background noise. The data are supposed to be obtained from different concentrations. This raises questions concerning the scientific soundness.  Please explain and check the manuscript data for further inconsistencies.

Author Response

We greatly acknowledge the reviewer for his constructive criticisms and very helpful insights, which contributed to improve the quality of the paper.

I am attaching an itemised reply letter to the comments made by the reviewer and changes have been introduced in the manuscript, accordingly to those comments.

All the modifications made in the manuscript have been highlighted.
